# X-ray Diffuse Scattering from Ca_3_NbGa_3_Si_2_O_14_ Single Crystal under External Electric Field Application

**DOI:** 10.3390/ma15217692

**Published:** 2022-11-01

**Authors:** Dmitrii Irzhak, Dmitry Roshchupkin

**Affiliations:** Institute of Microelectronics Technology and High-Purity Materials Russian Academy of Sciences, 142432 Chernogolovka, Russia

**Keywords:** X-ray diffuse scattering, piezoelectrics

## Abstract

X-ray diffuse scattering from the Ca_3_NbGa_3_Si_2_O_14_ (CNGS) crystal was measured with a triple axis X-ray diffractometer under the conditions of an external electric field. It is found that the nature of the intensity distribution of the asymmetrical part of diffuse scattering depends on the value of the applied electric field. This phenomenon is apparently associated with different piezoelectric characteristics of defect regions and the rest of the single crystal.

## 1. Introduction

X-ray diffuse scattering is a common method for studying the defective structure of single crystal materials. In an ideal crystal, the elastic scattering of waves by particles located at nodes of a periodic lattice due to interference occurs only for certain directions of the diffraction vector that coincide with the directions of the reciprocal lattice vectors. Ionic displacements from the lattice nodes violate the crystal periodicity, and the interference pattern changes; a smooth intensity component appears, corresponding to diffuse scattering. An analysis of the intensity distribution of diffuse scattering in the reciprocal space allows for the determination of the microdefect characteristics of the crystal structure [1,2,3,4].

The diffuse scattering method is widely used in the study of microdefects arising in crystals as a result of various influences, including implantation, radiation, annealing, etc. [5,6]. The diffuse scattering method is also used to study the effect of various growth parameters on the real structure of single crystals [7]. At the same time, studies of changes in a microdefect structure using the X-ray diffuse scattering method under the influence of sufficiently weak external actions, such as the external electric field application, propagation of acoustic waves, and elastic deformation, have not been carried out. This paper presents the results of a study of the effect of an external electric field applying on X-ray diffuse scattering in the Ca_3_NbGa_3_Si_2_O_14_ (CNGS) single crystal.

The previously studied sample of the CNGS crystal was used for measurements of independent piezoelectric moduli *d*_11_ and *d*_14_ [8]. It is found that the measured piezoelectric moduli are in good agreement with the results obtained on a crystal grown under the same growth conditions (an excessive Ga_2_O_3_ was added to the crystal growth raw material) [9], and differ by more than 15% from crystals obtained under the conditions of a stoichiometric compound with a mixture of N_2_ and 1–3 vol% O_2_ as used in the growth atmosphere [10,11]. Obviously, the difference in growth conditions affects the microdefect structure of crystals. This was demonstrated in [7] for single crystals of the lanthanum-gallium silicate group.

The study of the effect of an external electric field application on the intensity of X-ray diffuse scattering presented in this work can help to make an assumption about the influence of crystal structure microdefects on the piezoelectric properties of single crystals.

## 2. Experimental Set-Up

X-ray diffuse scattering on the CNGS crystal under external electric field application conditions was studied using a 4-circle Bruker D8 DISCOVER X-ray diffractometer (Bruker AXS Inc., Madison WI, USA) in the triple axis X-ray diffractometer scheme. The experimental set-up is schematically shown in Figure 1. X-ray radiation was collimated with a 100 µm input slit. The radiation source was an X-ray tube with a rotating copper anode ($CuK_{\alpha 1}$, λ = 1.54 Å). Ge (220) crystals with double reflection were used as the crystal-monochromator and crystal-analyzer.

X-cut plates with dimensions of 16×8×0.2 mm^3^ (length × width × thickness) for the crystals were used. The studied crystal was synthesized by performing the Czochralski method at the FOMOS Materials Co (Moscow, Russia). These crystals are non-ferroelectrics; therefore, the dependence of the strain on the magnitude of the electric field strength should be linear over a wide range of applied voltage values. To apply an external electric field, 100 nm thick A1-electrodes were formed on both sides of the single-crystal wafer using the thermal evaporation system for thin-film deposition. An electric field was applied along the $q_{z}$ direction.

The general map of the intensity distribution of diffracted X-ray radiation consists of the following two parts: the coherent one, to which the Bragg peak belongs, and the diffuse components. A detailed analysis of the effect of an external electric field applied to a piezoelectric crystal on coherently scattered X-ray radiation is presented in [8,12]. The distribution of the coherent part of the diffracted X-ray radiation in absolute angular coordinates should be used to calculate the piezoelectric modules. Relative coordinates in the reciprocal space are used to analyze diffuse scattering distribution maps. The transition from angular coordinates to the coordinates of the reciprocal space of the *q*, in accordance with [13], is carried out as follows:(1)$$\begin{array}{l} q_{x}=k\cdot (2\omega -\varepsilon )\cdot \sin\theta_{B};\\ q_{z}=-k\cdot \varepsilon \cdot \cos\theta_{B} \end{array}$$
where $q_{z}$, $q_{x}$ are components of the vector *q* in the perpendicular and parallel directions to the reciprocal lattice vector, respectively; and k=2π/λ is a wave vector of X-ray radiation with a wavelength λ; ω, ε are deviations of the crystal under study and the analyzer crystal from the exact value of the Bragg angle $\theta_{B}$.

The diffuse scattering intensity distribution map ($I_{\exp}^{d}\left(q\right)$) is divided into symmetric ($I_{sym}^{d}\left(q\right)$) and asymmetric ($I_{as}^{d}\left(q\right)$) components according to the following formulas: (2)$$\begin{array}{l} I_{sym}^{d}\left(q\right)=\frac{I_{\exp}^{d}\left(q\right)+I_{\exp}^{d}\left(-q\right)}{2}\\ I_{as}^{d}\left(q\right)=\frac{I_{\exp}^{d}\left(q\right)-I_{\exp}^{d}\left(-q\right)}{2} \end{array}$$
where *q* is a deviation of the scattering vector from the reciprocal lattice node.

The symmetrical component of diffuse scattering is determined by the symmetry of the displacement field around microdefects of the crystal structure. According to the theory of X-ray diffuse scattering on point crystalline defects and their clusters [14], the expansion of the contour in the *q*_x_ direction indicates a low symmetry of the field displacement due to the flat shape of defects; in the *q*_z_ direction, it means that the defect has a highly symmetric field displacement that is characteristic of defects with spherical shapes. The sign of the deformation of the crystal lattice Δ*V* caused by the defect is determined from the asymmetric part of X-ray diffuse scattering [15]. The deformation is positive (Δ*V* > 0) with *I_as_(q)* and a positive *q*_z_ > 0 for interstitial defects. For vacancy defects (Δ*V* < 0), *I_as_(q)* is positive for *q*_z_ < 0. 

## 3. Experimental Results

Figure 2 shows the maps of the asymmetric part of diffuse scattering for various reflections with an applied electric voltage of 0 V. The maps have a well-defined asymmetric structure. Diffuse scattering is shifted towards negative values of the *q*_z_ vector that indicates the presence of vacancy-type defects in the single-crystal wafer. 

Figure 3 shows the maps of the asymmetric part of diffuse scattering for various reflections with an applied electric voltage of −2000 V. Diffuse scatterings also have a well-defined asymmetric character with positive *I_as_(q)* values observed in the region of positive *q*_z_ values. Thus, the change in the volume of the crystal caused by the presence of a defect changes its sign; that is, the defect seems to have “turned” from a vacant to an interstitial one.

The positive voltage application of +2000 V does not cause a change in the nature of the intensity distribution of the asymmetric part of diffuse scattering in comparison with a voltage of 0 V. The corresponding distribution maps are shown in Figure 4. 

Figure 5 shows the symmetric parts of X-ray diffuse scattering on a CNGS crystal for reflection (220), obtained at various values of the electric voltage applied to the single crystal. On the maps, the intensity distribution is stretched along the *q*_z_ direction; therefore, microdefects are almost spherical in shape.

Obviously, the electric field application cannot affect the change in the microdefect symmetry to the extent that this is reflected in the distribution maps of the symmetric part of diffuse scattering. This is confirmed by the absence of significant differences on the maps of the symmetric part of diffuse scattering for various values of the electric voltage (Figure 5a–c).

Applying an electric field cannot change the defect type from vacancy to interstitial. Nevertheless, as can be seen from the distribution maps of the asymmetric part of diffuse scattering (Figure 2, Figure 3 and Figure 4), the deformation of the crystal lattice Δ*V* caused by the defect presence changes its sign from negative to positive when the applied voltage changes from 0 V to −2000 V but remains unchanged when the voltage changes in the opposite direction from 0 V to +2000 V. 

## 4. Discussion

Applying an electric field to a single crystal wafer causes compression or stretching of the crystal lattice that in turn causes the displacement of the coherent scattering peak in the *q*_z_ direction to the positive or negative side relative to the position of the Bragg peak of the undeformed crystal, depending on the direction of the applied electric field. Thus, the explanation of the observed effect may be that the electric field application causes different deformations in the defect region and in the rest of the single crystal matrix. If we assume that there are regions in the crystal in which the piezoelectric effect is absent or occurs less often, and the crystal lattice parameters of these regions differ from the matrix parameters, then, when the crystal matrix sizes change, the sign of deformation Δ*V* of these regions differs from the sign of deformation of the single crystal matrix in a positive or negative direction when changing the sign of the applied voltage. Figure 6 shows the schematic image of this phenomenon.

The presence of such defects can affect the magnitude of piezoelectric constants of the material. It can explain the differences in the measured values of piezoelectric coefficients of CNGS single crystals obtained under different growing conditions [8,9,10,11].

## 5. Conclusions

The effect of an applied external electric field on the character of diffuse scattering on a CNGS piezoelectric single crystal, found in the present work, is apparently associated with the presence in the single crystal volume of regions with piezoelectric characteristics different from the properties of a single crystal matrix. The presence of such regions certainly affects the piezoelectric characteristics of single crystals. Hence, to diagnose the quality of single crystal piezoelectric characteristics, methods of X-ray diffuse scattering should also be used in addition to X-ray coherent scattering (X-ray topography, X-ray diffractometry). 

## Figures and Tables

**Figure 1 materials-15-07692-f001:**
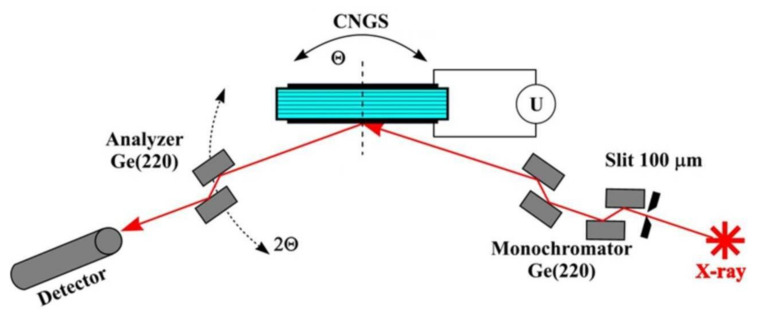
Triple axis x-ray diffractometer scheme for measurements of the independent piezoelectric moduli in the Bragg and Laue diffraction geometry.

**Figure 2 materials-15-07692-f002:**
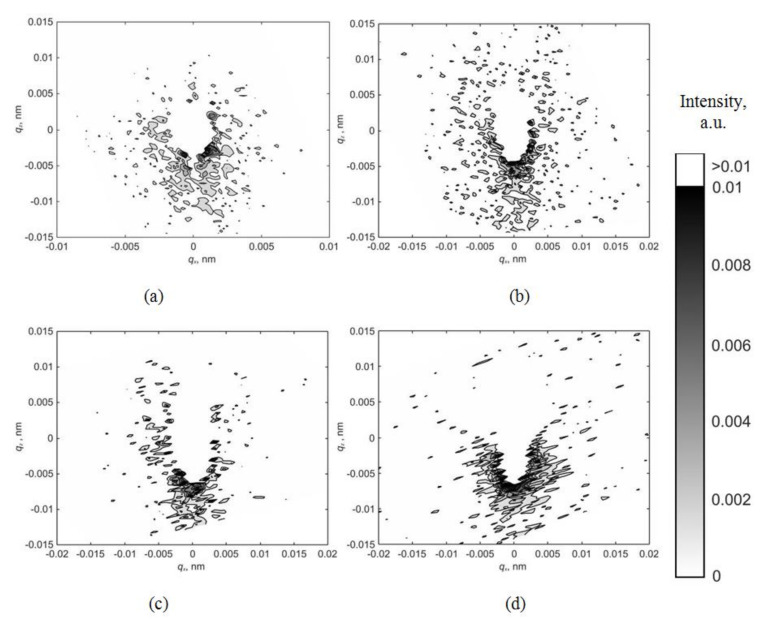
Maps of the asymmetric part of diffuse scattering for reflections (**a**) −(110); (**b**) −(220); (**c**) −(330); (**d**) −(440) at a voltage of 0 V.

**Figure 3 materials-15-07692-f003:**
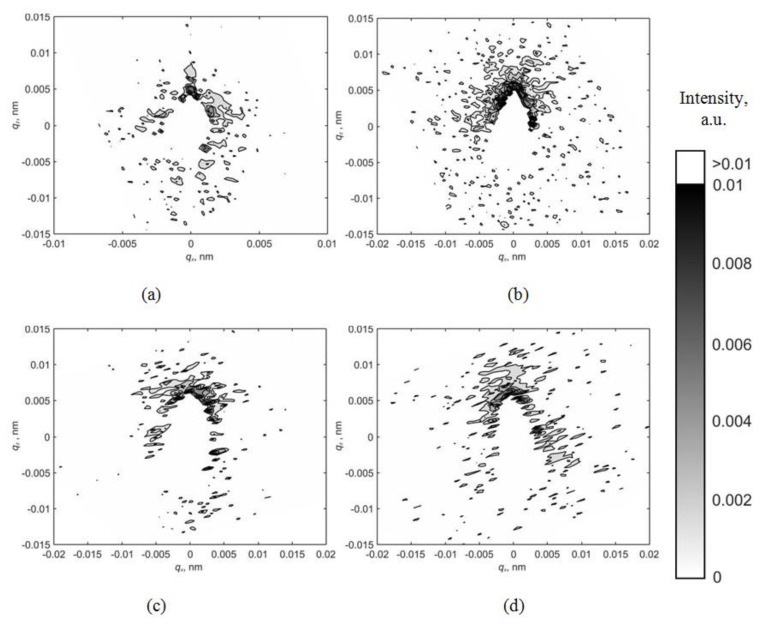
Maps of the asymmetric part of diffuse scattering for reflections (**a**) (110); (**b**) (220); (**c**) (330); (**d**) (440) at a voltage of −2000 V.

**Figure 4 materials-15-07692-f004:**
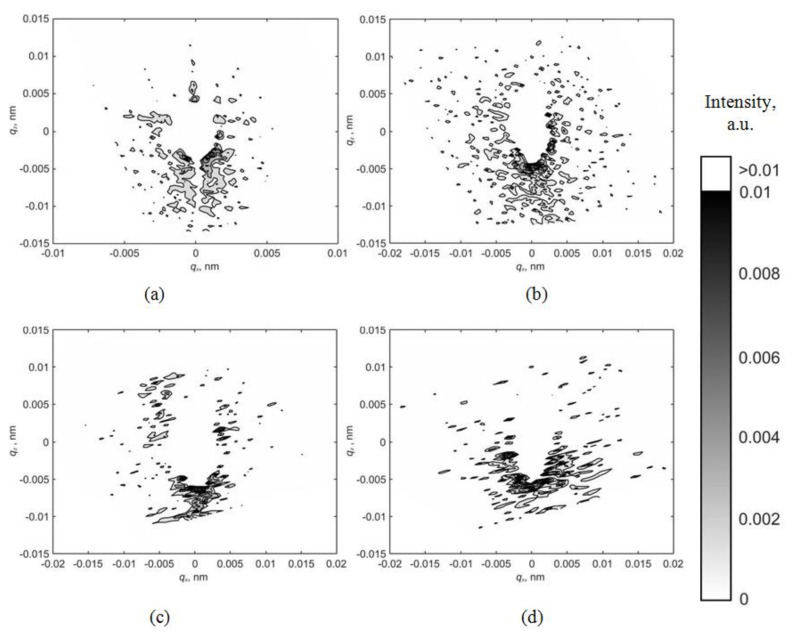
Maps of the asymmetric part of diffuse scattering for reflections (**a**) (110); (**b**) (220); (**c**) (330); (**d**) −(440) at a voltage of +2000 V.

**Figure 5 materials-15-07692-f005:**
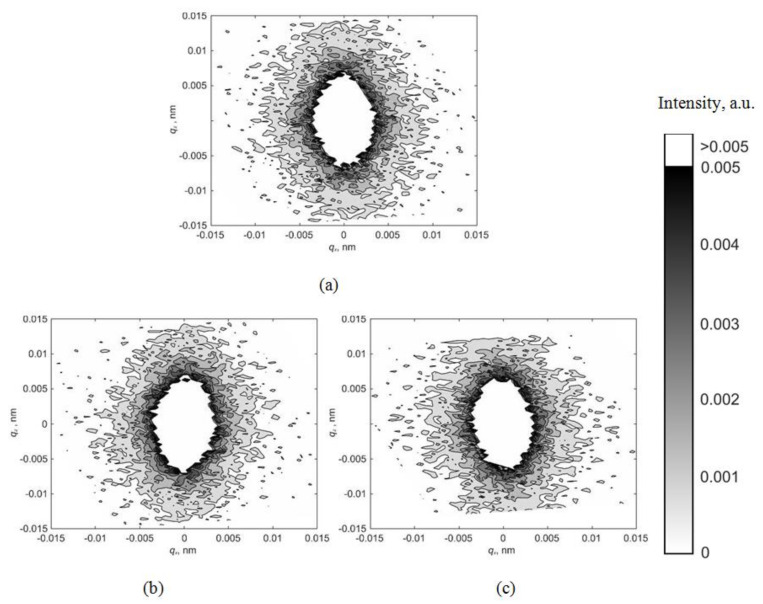
Maps of the symmetric part of diffuse scattering for reflections (220) at various values of applied electric voltage (**a**) 0 V; (**b**) −2000 V; (**c**) +2000 V.

**Figure 6 materials-15-07692-f006:**
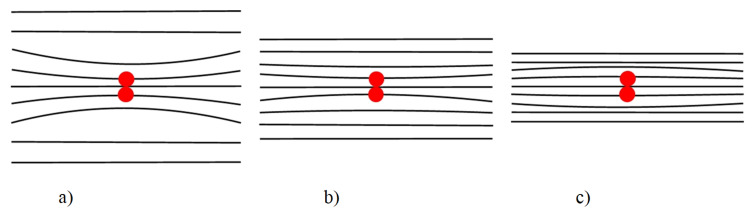
Schematic image of the effect of an electric voltage application on changing the sign of the crystal lattice deformation Δ*V* caused by the presence of a defect (red spot): (**a**) with an increase in the crystal lattice parameters of the matrix (positive voltage application under the conditions of the described experiment) Δ*V* < 0; (**b**) in the original crystal without an electric field application Δ*V* < 0; (**c**) a decrease in the lattice parameters of the matrix when a negative electric voltage is applied Δ*V* > 0.

## Data Availability

Data available on request from the authors.

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
