# Peer review of "X-ray Diffuse Scattering from Ca_3_NbGa_3_Si_2_O_14_ Single Crystal under External Electric Field Application"

_materials, 2022, doi:10.3390/ma15217692_

Round 1
Reviewer 1 Report
The authors investigate the diffuse scattering of single crystals of the piezoelectric compound Ca3NbGa3Si2O14. They find changes in the nature of the antisymmetric diffuse scattering under an applied electric field, which they attribute to differences in the intrinsic piezoelectric response and the piezoelectric response of defect regions. The paper is clearly written and the results are novel.
The authors should address the following minor points:
(1) Line 44: micridefects -> microdefects
(2) Lines 79-82: I don’t understand whether this is a general statement or a result for this particular system.
(3) Line 85: “defects between points”. Do you mean “interstitial defects” ?
(4) Line 93: What is the relationship between electric field direction and “q” direction?
(5) Line 99: How strong is the electric field in V/m ?
(6) Line 124: less occurs -> occurs less often
(7) Figure 6.: You should explicitly note that such asymmetry is possible because the crystal structure is noncentrosymmorphic.
(8) References: The reference list is short. You should add more citations to previous work on this compound.
Author Response
(1) Line 44: micridefects -> microdefects
fixed
(2) Lines 79-82: I don’t understand whether this is a general statement or a result for this particular system.
It is a general statement. Reference added.
(3) Line 85: “defects between points”. Do you mean “interstitial defects” ?
Yes.
(4) Line 93: What is the relationship between electric field direction and “q” direction?
Electric field was applied along qz direction. Added to "experimental set-up"
(5) Line 99: How strong is the electric field in V/m ?
It is easy to calculate: 2000V/0.2mm=10kV/mm or 10 000 000 V/m
(6) Line 124: less occurs -> occurs less often
(7) Figure 6.: You should explicitly note that such asymmetry is possible because the crystal structure is noncentrosymmorphic.
This is obvious since the article is about a piezoelectric crystal. And as you know, the piezoelectric effect is observed only in non-centrosymmetric crystals.
(8) References: The reference list is short. You should add more citations to previous work on this compound.
I think it is enough references for the reader who is really interested in the topic of the paper.
Reviewer 2 Report
· Few references
· No degree of novelty
· Only X-ray diffraction is not enough, the work should contain more experimental analyses
Author Response
I will be glad if you demonstrate me any paper about X-ray diffuse scattering under external electric field application.
Reviewer 3 Report
Dear Editor,
I have read the manuscript entitled: “X-ray diffuse scattering from Ca3NbGa3Si2O14 single crystal under external electric field application” and I would like to address following suggestions to the authors:
1-please explain why you used this Ca3NbGa3Si2O14 (CNGS) crustal?
2-Abstract and conclusion should contain more quantitative information.
3-Please more explain about the novelty and application of this study.
4-In figures 2, 3, 4, 5 how to be sure about (hkl)? Please put XRD pattern (if available)
5- The introduction can be improved by providing a more critical discussion of recent related literature. For example, some papers related such as: Materials Science in Semiconductor Processing, 36, 134-139, (2015); Investigation of The Electrical Resistivity Of 20μm-Gap Gold-DNA-Gold Structure: Exploiting the Current-Voltage Characteristics Under a Variable External Magnetic Field. in Applied Mechanics and Materials (2014); Sensors, 14, 19229-19241, (2014); Current-voltage characterization on Au-DNA-Au junctions under the influence of magnetic field. in Advanced Materials Research. 2012 should be cited.

Author Response
1-please explain why you used this Ca3NbGa3Si2O14 (CNGS) crustal?
The article presents a part of the results carried out as part of a large project to study the characteristics of a crystal. The main results have been published previously and are mentioned in the reference list.
2-Abstract and conclusion should contain more quantitative information.
When studying diffuse scattering, quantitative data can be obtained using simulations. However, for this you need to have some kind of model. In this case, even the coincidence of the experimental results with the simulation results does not guarantee that the model and simulation parameters correspond to the true nature. This paper presents experimental results, and modeling was not the goal of the research.
3-Please more explain about the novelty and application of this study.
I have not seen any papers that describe the change in the asymmetry of diffuse scattering maps under the influence of an electric field. Therefore, I consider it difficult to add any description of novelty, because there is nothing to compare with.
4-In figures 2, 3, 4, 5 how to be sure about (hkl)? Please put XRD pattern (if available)
This is a single crystal plate cut parallel to certain crystal planes. Therefore, any symmetric reflection is uniquely determined.
5- The introduction can be improved by providing a more critical discussion of recent related literature. For example, some papers related such as:
Materials Science in Semiconductor Processing, 36, 134-139, (2015);
Investigation of The Electrical Resistivity Of 20μm-Gap Gold-DNA-Gold Structure: Exploiting the Current-Voltage Characteristics Under a Variable External Magnetic Field. in Applied Mechanics and Materials (2014); Sensors, 14, 19229-19241, (2014);
Current-voltage characterization on Au-DNA-Au junctions under the influence of magnetic field. in Advanced Materials Research. 2012 should be cited.
I don't understand what connection there is between the change in current-voltage characteristics when applying a magnetic field and the results presented in my paper.
Round 2
Reviewer 2 Report
Paper was improved.
Reviewer 3 Report
Dear Editor,
I ask the authors to discuss the results obtained in an honest way. Accordingly, I believe the current manuscript does not contain enough novelty and content to be published in materials journal.